# Energy Metabolic Disorder of Astrocytes May Be an Inducer of Migraine Attack

**DOI:** 10.3390/brainsci12070844

**Published:** 2022-06-28

**Authors:** Junhua Li, Xiaotong Ye, Yang Zhou, Shiqiao Peng, Peibing Zheng, Xiaoxiao Zhang, Jiajun Yang, Yanhong Xu

**Affiliations:** 1Central Laboratory, Shanghai Jiao Tong University Affiliated Sixth People’s Hospital, Shanghai 201306, China; junhualijh@163.com (J.L.); clearling405@163.com (S.P.); 2National Demonstration Center for Experimental Fisheries Science Education, Shanghai Ocean University, Shanghai 201306, China; xiaotongye96@163.com (X.Y.); zhouyang109255@163.com (Y.Z.); 3Neurology Department, Shanghai Jiao Tong University Affiliated Sixth People’s Hospital, Shanghai 201306, China; peibingzheng_92@163.com (P.Z.); 14211290004@fudan.edu.cn (X.Z.); 4Neurology Department, Shanghai Sixth People’s Hospital East Campus, Shanghai University of Medicine & Health Sciences, Shanghai 201306, China

**Keywords:** migraine, fasting, mitochondrial dysfunction, inflammatory factor, reactive oxygen species (ROS)

## Abstract

Migraine is a chronic headache disease, which ranks second in years lost due to disability. However, the mechanism of migraines is still not clear. In migraine patients, fasting can trigger headache attacks. We explored the probable mechanism of why fasting can induce headaches. Nitroglycerin (NTG) was used to induce acute migraine attacks in mice. Primary astrocytes were used to study the pathophysiological mechanism and a Seahorse analyzer was used to detect mitochondrial function. NTG induced more serious headaches in the fasting group. Both the head-scratching times and climbing-cage times in the fasting group were higher than those in normal-diet group. More ROS and inflammatory factors, such as IL-6 and IL-1β, were induced in low-glucose conditions. Seahorse showed that the basal oxygen consumption rate (OCR) and OCR for ATP production were lower in mice who had received NTG with low glucose levels than in other groups. The activity of AMPK was inhibited in this group, which may explain the Seahorse results. We concluded that in the low-glucose state, astrocytes produce more inflammatory factors, ROS, which may be a result of mitochondrial metabolism dysfunction. Improving mitochondrial function and supplying enough substrates may be an option for relieving migraine attacks.

## 1. Introduction

Migraine is a chronic neurological disorder characterized by attacks of moderate to severe headaches and reversible neurological and systemic symptoms [1]. According to the 2016 Global Burden of Disease study, migraine ranks second in years lost due to disability, representing 5.6% of all years lost due to disability globally [2,3]. It affects people of all ages, and is the most prevalent between the ages of 25 and 55 years.

However, the pathogenetic mechanism of migraines is still unclear. Factors such as genetics, diet, mental state, mitochondrial dysfunction and so on are believed to participate in the pathogenesis of migraines. Abnormal vasomotion, inflammation, reactive oxygen species (ROS) and neurotransmitter disorders are related to the onset of migraines. In migraine patients, fasting can trigger migraine attacks [4,5]. The number of migraine days and severity of headaches increased in Muslim migraine patients fasting during Ramadan (fasting for a month) [5,6]. It is hypothesized that the imbalance between the excitatory and inhibitory terminals, causing the collective depolarization of neurons and astrocytes in a network, may trigger auras and/or headaches in the fasting state [4]. Nevertheless, the mechanism of why fasting triggers headache still remains poorly understood.

Energy metabolic impairment is involved in the pathophysiology of migraine, as shown by biochemical, morphological and magnetic resonance spectroscopy (MRS) studies [7]. Mitochondrial oxidative phosphorylation is the main source of cellular energy. Migraine is one of the clinical characteristics of mitochondrial disorders, such as mitochondrial encephalomyopathy with lactic acidosis and stroke-like episodes (MELAS) [8]. Impaired mitochondria produce excessive ROS, which may mediate cortical spreading depression (CSD) [9]. The inflammation triggered by oxidative stress is the cause of many chronic diseases, including migraine [10]. Neuroinflammation is an important aspect in the pathophysiology of migraine [11,12].

Astrocytes (AS), a type of glial cell, have emerged as active players in brain energy delivery, production, utilization, and storage [13]. AS always accompany the process of neuron development in the central nervous system. For a long time, it has been thought that astrocytes only act as scaffolds for neurons. However, in recent years, research has suggested that astrocytes play an important role in the development, normal physiology and pathological processes of the central nervous system [14,15]. In addition to supporting and isolating effects, astrocytes can also regulate the concentration of ions inside and outside nerve cells, transmit second messengers, and absorb, inactivate or supply neurotransmitters [14]. The role of astrocytes in craniocerebral trauma, neurodegenerative diseases and epilepsy has been studied [16,17,18]. However, there have been few studies examining the role of astrocytes in the field of migraine.

In this study, we aimed to explore the pathophysiology mechanism of energy deficit in migraine attack, focusing on mitochondrial function, oxidative stress and inflammation in astrocytes.

## 2. Materials and Methods

### 2.1. Animals and Behavior Testing

All behavioral studies were conducted on male C57BL/6 mice (aged 10–12 weeks, SPF grade) which were provided by Shanghai Xipu’er Bikai Laboratory Animal Co., Ltd. (Shanghai, China). Only male mice were used so as to avoid the influence of physiological cycle and sex hormones of female mice. Mice were maintained at a temperature of 22 ± 2 °C with a humidity of 50 ± 5% on a 12 h light–dark cycle; food and water were freely available. All procedures involving the use of animals were approved by the Institutional Animal Care and Use Committee (IACUC) of the Shanghai Sixth People’s Hospital affiliated with the Shanghai Jiao Tong University. NTG was freshly diluted in 0.9% saline and a dose of 10 mg/kg was given subcutaneously to induce acute migraine model. A cohort of 12 mice from each group, fasting for 16 h or receiving normal food, were observed 120 min after NTG injection. Each mouse was placed separately in a transparent apparatus after NTG injection, and the behavior of mice was recorded with a video camera (SONY, HDR-CX150E, Tokyo, Japan). To evaluate the headache behavior, the number of instances of head scratching and cage climbing of each mouse were subsequently recorded in 3 consecutive time periods: 0–30, 30–60, and 60–90 min after NTG injection. 

### 2.2. Primary Astrocytes Isolation and Culture

Neonatal 1- to 3-day-old C57BL/6 suckling mice purchased from Shanghai Xipu’er Bikai Laboratory Animal Co. Ltd. were anesthetized with sevoflurane and disinfected with 75% alcohol. The mice were then scalped and their skulls were opened, and the brains were collected and transferred into chilled Dulbecco’s Modified Eagle Medium (DMEM, SH30243.01, HyClone, Los Angeles, CA, USA). Under stereomicroscopic observation, the meninges, olfactory bulb, cerebellum and brainstem were removed. The cerebral hemisphere was cut into fragments, and then digested with 0.025% trypsin. Complete medium (high glucose DMEM containing 10% FBS, 100 units/mL penicillin, 100 μg/mL streptomycin) was used to terminate digestion. The cell suspension was filtered with 200 mesh, and then centrifugated at 1000 rpm for 5 min. Cell precipitate was resuspended with complete medium, and cultured in 5% CO_2_, in a 37 °C incubator. When the cells filled the culture flask, they were replaced with new medium and shaken at 100 rpm for 6 h to remove microglia. Cells were used in the experiment after 2–3 passages.

### 2.3. ROS Assay and Mitochondrial Membrane Potential Test

ROS production of NTG-treated astrocytes in different glucose conditions was quantified with a DCFH-DA (2,7-Dichlorodi-hydrofluoresceindiacetate) probe in the Reactive Oxygen Species Assay Kit (S0033S, Beyotime, Shanghai, China). The DCFH-DA treatment was conducted at 10 μM under 37 °C for 30 min in the absence of light. The green fluorescence intensity was determined with flow cytometry (FACSCelesta, BD, Franklin Lake, NJ, USA). 

Mitochondrial membrane potential (MMP) was detected with JC-1 (10 μg/mL, 40705ES08, YEASEN, Shanghai, China) to evaluate the mitochondrial function. The method was same as described elsewhere [19]. The red fluorescence signal in healthy mitochondria and the green fluorescence signal of damaged mitochondrial potential were detected with flow cytometry in PE and FITC channels, respectively. The ratio of PE/FITC was calculated in each sample to determine the MMP.

### 2.4. Mitochondrial Function Assay

The mitochondrial function assay was conducted by measuring the oxygen consumption rate (OCR) in cells with the Seahorse XFe24 equipment (Seahorse Bioscience, Agilent, Santa Clara, CA, USA). This procedure is the same as previously described [20]. Astrocytes were seeded in the 24-well Seahorse plate at 5 × 10^4^ cells/well 24 h before the test. The medium changed to normal high-glucose (4.5 g/L, same as culture medium) or low-glucose (1 g/L) Base Medium (102353-100, Agilent, Santa Clara, CA, USA) and NTG was given at 2 μg/mL 1 h before the test. The activators and inhibitors were: Oligomycin (1 μM, 1404-19-9-C, Sigma, Saint Louis, MO, USA), FCCP (2 μM, 370-86-5, Sigma), Antimycin A (0.5 μM, 2247-10, Biovision, Waltham, MA, USA), and Rotenone (0.5 μM, R8875, Sigma). Basal mitochondrial OCR, proton leak and ATP production were calculated based on the changes in OCR under the corresponding conditions.

### 2.5. Quantitative Real-Time Polymerase Chain Reaction (qRT-PCR) Assay

Total RNA was extracted from astrocytes using total RNA extraction reagent (R401-01, Vazyme Biotech, Nanjing, China). An amount of 1 μg of mRNA was used to generate cDNA by using a first-strand cDNA synthesis kit (HiScript^®^ II Q RT SuperMix, R223, Vazyme Biotech, Nanjing, China). The primer sequences were synthesized as shown in Table 1. The results were normalized at the 18s mRNA level.

### 2.6. Western Blot (WB)

Protein levels of interest were determined by Western blot analysis according to a protocol described elsewhere [20]. Cells were treated with NTG in the 6-well plate, and the whole-cell lysates were quantified with a BCA kit (P0012, Beyotime, Shanghai, China). The protein sample (30 μg/sample) was loaded in each lane of the gel. The primary antibodies to pAMPK (α1T183 + α2T172, ab133448), pPDE1a (Ser297, ab177461), Citrate synthase (ab129095), IDH2 (ab131263), MDH2 (ab181873), Complex I (NDUFB8, ab110242), Complex II (ab110410), Complex III (UQCRC2, ab14745), Complex IV (COX IV, ab16056) and β-tubulin (ab6046) were obtained from Abcam (Cambridge, MA, USA). Antibody to ATP5A1 (495240) was from Invitrogen (Camarillo, CA, USA). β-tubulin was used as the internal control. The signal was collected with a Chemiluminescence gel imager (Amersham Imager 600, GE Healthcare, Chicago, IL, USA) and the images were quantified with the Image J software.

### 2.7. Statistical Analysis

All experiments were repeated three times with consistent results. GraphPad Prism 8.0.1 (GraphPad Software, San Diego, CA, USA) was used in the statistical analysis. The data were tested to be normal distribution, and was statistically analyzed with the Student T-test. The data were presented as mean ± SEM with a significance of *p* < 0.05.

## 3. Results

### 3.1. Headache Caused by NTG Is more Serious in Fasting Group Mice

NTG is commonly used to prepare migraine models. As the symptoms of headache, the head-scratching and cage-climbing times of the migraine model mice were observed in 0–30, 30–60, and 60–90 min intervals after NTG injection. The mean head-scratching times of the fasting mice were increased compared to the control mice at different time intervals (*p <* 0.05) (Figure 1A). The mean climbing-cage times were also increased in the fasting group compared to control mice at different time intervals, although the statistical difference was not significant (*p* = 0.073) (Figure 1B).

### 3.2. More ROS Is Induced by NTG in Low-Glucose Condition in Astrocytes

Accumulating evidence has demonstrated a possible role of mitochondrial dysfunction in migraine, which is related to an increased production of ROS [21]. To study the role of NTG in the mitochondrial function of astrocytes, we examined ROS production in astrocytes in different glucose conditions with NTG treatment for 1 h. As we expected, NTG induced ROS production under both the low- and normal-glucose conditions. The quantitative analysis indicated that the ROS level in low-glucose condition with NTG was about 2-fold higher than normal glucose with NTG treatment (Figure 2A,B). The mitochondrial membrane potential did not change at different glucose levels with NTG treatment (Figure 2C). 

### 3.3. More Inflammatory Factors Are Produced in Low-Glucose Group

It is well recognized that neuroinflammation is involved in the pathogenesis of various neurological disorders, including migraine. Primary astrocytes were collected after treatment in different glucose with or without NTG for 4 h, to explore inflammatory cytokine production. qRT-PCR showed that the mRNA levels of pro-inflammatory factors IL-1β, IL-6 and IL-1Ra were strikingly increased in low glucose, especially with NTG treatment (Figure 3A–C). Low glucose and NTG had an additive effect on the production of IL-6 and IL-1Ra. 

### 3.4. NTG Reduces ATP Production in Low-Glucose Conditions by Inhibiting AMPK Activity

To understand the NTG activity in mitochondrial metabolism, we employed Seahorse analyzer and detected the OCR of the primary astrocytes with NTG in different glucose conditions. The results suggested that in response to NTG treatment, astrocytes with low glucose showed clearly decreased OCR levels at baseline, but OCR did not change under low glucose without NTG. OCR is composed of two parts, ATP production and proton leakage. The analyzed data revealed that less ATP production, caused by NTG in low-glucose conditions, mainly contributed to the decreased level of OCR, compared with the unchanged proton leakage level (Figure 4A).

To further study the effects of NTG on mitochondrial metabolism during starvation, the mitochondria-related proteins of primary astrocytes were investigated. The Western blot assay suggested that NTG did not affect mitochondrial respiratory chain complexes I-IV and ATP5A protein levels (Figure 4B). The local metabolic enzymes in tricarboxylic acid cycle, such as citrate synthase (CS), isocitrate dehydrogenase 2 (IDH2), and malate dehydrogenase 2 (MDH2), also had no noteworthy change with NTG treatment, neither in the high-glucose nor in low-glucose conditions (Figure 4C). 

AMP-activated protein kinase (AMPK) is activated when the ratio of AMP and ATP is elevated, which means an insufficient energy states and regulates energy homeostasis [22]. The phosphorylation state is the activated form. In the test of Seahorse, we observed the decreased level of ATP production in the low-glucose group with NTG. However, in the WB test, we did not see an activation of AMPK, but a decreased phosphorylation level of AMPK was observed (Figure 4B). So, the inactivation of AMPK may be the reason for the decreased ATP production in the low-glucose condition with NTG. 

## 4. Discussion

Migraine is a chronic disorder, with a complex pathophysiology involving neuronal and vascular mechanisms [23]. More and more studies indicate that migraine is a metabolic disease and associated with mitochondrial dysfunction [24,25]. Migraine is associated with stroke [26,27], which may reflect mitochondrial dysfunction. Brain white matter lesions are frequently detected in migraine patients and the frequency of migraine is not associated with these lesions [28,29]. The lesions of brain white matter may be a result of insufficient energy in the brain. Study results of 31P-MRS suggest a reduced availability of neuronal energy and imply a mitochondrial dysfunction in the migraine brain [30]. 1H-MRS studies showed that N-acetyl aspartate levels were decreased in migraine, probably due to mitochondrial dysfunction and abnormal energy metabolism [30,31]. The classic mitochondrial disease, MELAS, also presents with a migraine-like headache [32], which is further evidence that mitochondria and metabolism are linked in migraine attacks. This evidence indicates that migraine is a disease related to mitochondrial dysfunction. In this study, fasting and low glucose were used to simulate energy deficiency, and a Seahorse analyzer was used to detect astrocyte mitochondrial function directly. Our results revealed that the mitochondrial dysfunction of astrocytes play a role in the pathogenesis of migraine attacks.

Fasting triggers headache attacks in migraine patients, even in patients without migraine [4]. Hypoglycemia is the first reasonable possibility as a trigger for a migraine attack, although there might be some other reasons, such as dehydration or the activation of the sympathetic nervous system [4,5]. A study showed a decrease in active migraine prevalence prior to diabetes diagnosis, and a lower risk of developing type 2 diabetes for women with active migraines [33]. So, low glucose levels or energy insufficiency may act as an intermediary of headache attacks in migraine patients. Hypoglycemia may trigger migraine attacks through the oxidative stress mechanism [34]. 

NTG is a commonly used drug for migraine modeling [35]. In addition to releasing nitric oxide to cause vasodilation, it also has other effects, such as inducing the production of superoxide and peroxynitrite radicals [34]. In our study, we used NTG to induce headache in mice and detected the influence of NTG on astrocytes, which are important in brain energy metabolism [13]. We found that the headaches of the fasting group were more serious than in the normal-diet group, which is consistent with clinical studies. This may be due to insufficient ATP, the presence of more inflammatory factors and ROS production.

OCR detected by Seahorse is a reflection of mitochondrial function [36]. In our study, we detected the OCR of astrocytes in different conditions, and found low glucose or NTG alone did not result in a significant influence of OCR. However, an obvious decrease in OCR was observed in the low-glucose condition with NTG. The OCR for ATP production decreased, which is a sign of insufficient energy. AMPK is thought to be a nutrient and energy sensor that maintains energy homeostasis and is activated to promote ATP production when energy is insufficient [37]. However, in this study, the activity of AMPK in the decreased ATP group (low glucose with NTG) was not activated, but reduced. So, the decreased activity of AMPK may be the reason reduced ATP production was observed. 

The lack of energy in astrocytes induced more inflammatory factors, even without NTG, which may explain low glucose levels triggering headaches in some patients without migraines. Several major cytokines, including IL-1β, IL-6 and tumor necrosis factor (TNF), have been thought to be linked to migraine pathophysiology, as their levels are altered in individuals with migraine [38]. CSD is a propagating wave of profound depolarization in cerebral gray matter [39], and is related to migraine and other neurological diseases such as epilepsy and cerebrovascular diseases [40,41,42]. Astrocytes mediate inflammation in CSD by elevating the expression of the pro-inflammatory markers IL-6, IL-1β, and TNFα [39]. In our study, we found that in low-glucose conditions, the pro-inflammatory markers IL-6 and IL-1β were elevated, especially with the induction of NTG, which is consistent with the literature. However, the anti-inflammatory factor IL-1Ra was also elevated in our study. Coenzyme Q10 (CoQ10), which can improve mitochondrial function, is also an anti-inflammatory agent. In a clinical study, CoQ10 significantly improved the frequency, severity and duration of migraine attacks [43]. We did not verify if CoQ10 can reverse the reduced OCR under low glucose with NTG in our system. This is one of the limitations of our research. Nevertheless, we confirmed that energy deficiency is an inducer of inflammation in astrocytes.

ROS, which are produced due to abnormal mitochondria function, are related to many neurological diseases, such as stroke, Parkinson’s disease, Alzheimer’s disease and so on [44,45,46]. Migraine sufferers exhibit impaired metabolic capacity, with an increased formation of ROS [21]. In our study, NTG induced more ROS in the low-glucose condition than in the high-glucose condition. The mitochondrial dysfunction of this group, detected by Seahorse, may explain the elevation of ROS. Inhibiting ROS may have therapeutic benefits in preventing migraines [9]. Antioxidants may become a potential drug for migraine treatment.

In our study, we found decreased AMPK activity in the NTG group with low glucose, but not in the NTG group or low glucose group alone. This may be a result of decompensation. In addition, we only used astrocytes as the cell models in our system. Do the interactions between astrocytes and neurons or microglia play a role in the attack of migraines induced by fasting? Further exploration is required.

## 5. Conclusions

In low-glucose states, astrocytes produce more inflammatory factors, ROS, which may be a result of the dysfunction of mitochondrial metabolism. Improving mitochondrial function, supplying enough energy metabolism substrates or antioxidants may be options for relieving migraine attacks. Of course, basic experiments are still needed to indicate the exact mechanisms in the pathophysiology of migraine and clinical trials for improving mitochondrial function and supplying sufficient energy substrates and antioxidants are also essential to evaluate their effectiveness in reducing migraine attacks.

## Figures and Tables

**Figure 1 brainsci-12-00844-f001:**
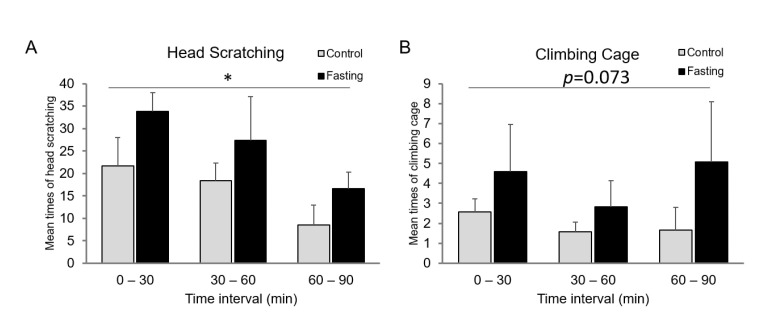
Headache caused by NTG is more serious in fasting group. (**A**) Mean times of head scratching were increased in the fasting group compared to the control group at different time intervals. (**B**) Mean cage-climbing times of mice were increased in the fasting group compared to the control group at different time intervals. NTG, nitroglycerin, 10 mg/kg, subcutaneous injection. The fasting group mice were starved for 16 h. *n* = 12. * *p* < 0.05.

**Figure 2 brainsci-12-00844-f002:**
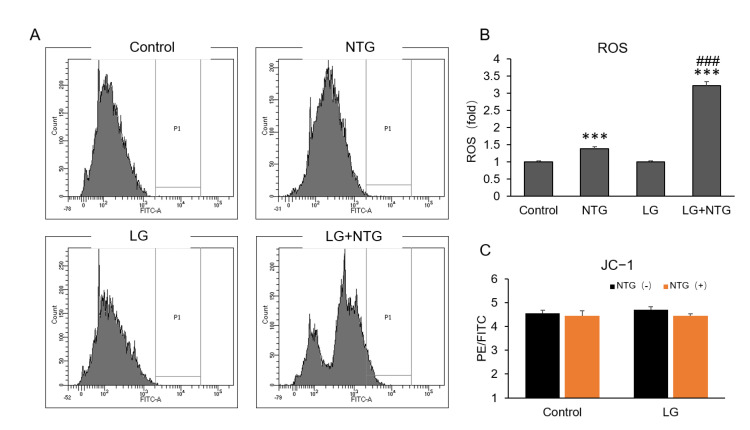
More ROS are induced by NTG in low-glucose condition in astrocytes. (**A**) ROS of astrocytes were detected by flow cytometry after NTG treatment at a dose of 2 μg/mL for 1 h. Control, glucose 4.5 g/L; low glucose (LG, glucose 1 g/L). (**B**) Quantitative analysis showed that ROS level in low-glucose condition was about 2-fold compared with normal glucose after treated by NTG. (**C**) Mitochondrial membrane potential detected by JC−1 showed no change in different conditions. ROS, reactive oxygen species ***, *p* < 0.001, compared with control; ###, *p* < 0.001, compared with NTG.

**Figure 3 brainsci-12-00844-f003:**
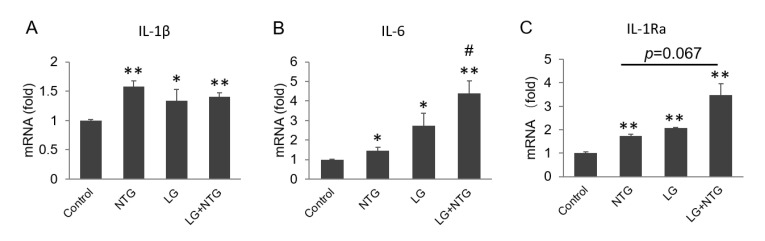
More inflammatory factors are produced in low-glucose group. Astrocytes were collected after treatment with different concentrations of glucose and NTG (2 μg/mL) for 4 h. mRNA levels of IL-1β, IL-6 and IL-1R were detected by qRT-PCR. (**A**) Both NTG and low glucose induced IL-1β production. (**B**) IL-6 was induced by NTG and low glucose. (**C**) IL-1Ra was induced by NTG and low glucose. Compared with control, * *p* < 0.05, ** *p* < 0.01. Compared with NTG, # *p* < 0.05.

**Figure 4 brainsci-12-00844-f004:**
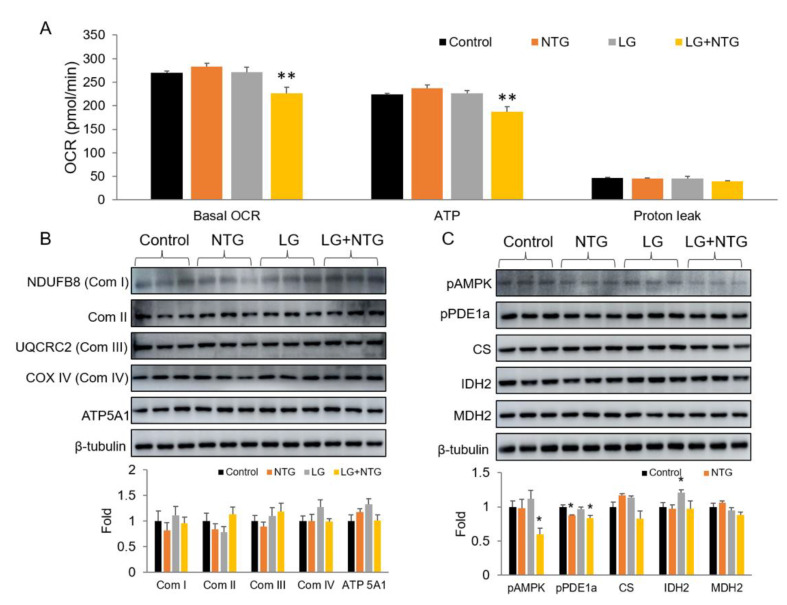
NTG reduced ATP production in low glucose by inhibiting AMPK activity. (**A**) Seahorse was used to detect oxygen consumption rate (OCR). A total of 5 × 10^4^ astrocytes were seeded in a well of 24-well plate used in XFe24 Seahorse analyzer. *n* = 5. Basal OCR and OCR for ATP production and proton leak were calculated with the instruction of Seahorse. (**B**) Proteins of mitochondrial complexes underwent no significant change after the treatment of NTG, neither in high-glucose nor in low-glucose conditions. (**C**) Protein level of pAMPK reduced significantly in the low-glucose and NTG group. NTG 2 μg/mL, treated for 4 h. AMPK, AMP-activated protein kinase; CS, citrate synthase; IDH, isocitrate dehydrogenase 2; MDH2, malate dehydrogenase 2. Compared with control, * *p* < 0.05, ** *p* < 0.01.

**Table 1 brainsci-12-00844-t001:** Primers used in qRT-PCR.

Name	Forward (5′-3′)	Reverse (5′-3′)
IL-1β	GGACAAGCTGAGGAAGATGC	TGGAGAACACCACTTGTTGC
IL-6	CTCCCAACAGACCTGTCTATAC	CCATTGCACAACTCTTTTCTCA
IL-1Ra	TTGTGCCAAGTCTGGAGATG	CTCAGAGCGGATGAAGGTAAAG
18s	GCCGCTAGAGGTGAAATTCT	TCGGAACTACGACGGTATCT

## Data Availability

The data presented in this study are available on request from the corresponding author.

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
