# Peer review of "Energy Metabolic Disorder of Astrocytes May Be an Inducer of Migraine Attack"

_brainsci, 2022, doi:10.3390/brainsci12070844_

Round 1

Reviewer 1 Report

Dear Authors, 

In the manuscript, the authors examined the effect of fasting in migraine. In their experiment, they used the NTG model to induce migraine-like attacks in mice. Furthermore, primary astrocytes were used to study the pathophysiological mechanism, and a Seahorse analyzer was used to detect mitochondrial functions. NTG was found to induce more severe headache attacks in fasting animals, and more ROS and inflammatory factors such as IL-6 and IL-1β were induced in low glucose conditions. In addition, basal oxygen consumption and ATP production were lower in NTG with low glucose than in other groups. Based on their results Improving mitochondrial function and supplying enough substrates may be a good choice for relieving migraine attacks. 

The topic is timely and may attract much attention. However, in its current version, the manuscript has several limitations that should be addressed.

I have some suggestions to improve this paper:

1. In general, I recommend authors use more references to back their claims, especially in the Introduction of the article, which I believe is lacking. Thus, I recommend the authors attempt to deepen the subject of their article, as the bibliography is too concise. Nonetheless, in my opinion, less than 50 articles for a research paper are insufficient. Currently, authors cite only 33 papers, and in my opinion, they should cite more than it. Therefore, I suggest the authors focus their efforts on researching relevant literature: I believe that adding more citations will help to provide better and more accurate background to this study.

2. Introduction:

Write in more detail about the relationship between migraine and fasting/starvation. Why was this investigated?

In addition, it would be worthwhile to write about the role of mitochondrial dysfunction in migraine.

Furthermore, the relationship between inflammatory processes and migraine should be detailed.

At the end of the introduction, do not detail / discuss their results, but write about the aims of their experiment, why they examined the relationship between migraine and fasting, why did they do these studies?

3. Materials and Methods

In animal experiments, if the behavior of the animals is examined, the minimum number of animals required is 8-10. This number of animals is essential to carry the appropriate statistical test.

4. Figures

Please include the full form of the abbreviations in the figure legends as well.

Figure 1: please include SEM or SD on the graph. 

Figure 3C: n=0.067 - i think they wanted to indicate the value of p. Please check it. 

5. Discussion 

The discussion is unfocused. Only consecutively described sentences, the discussion is not structured properly. 

From the summary, I miss that the authors detail the importance of their results, and it would be good to summarize how their findings relate to knowledge about this disease. 

6. Language proofreading is recommended.

Recommendation of revision: major

Author Response

Response to Reviewer 1

  1. In general, I recommend authors use more references to back their claims, especially in the Introduction of the article, which I believe is lacking. Thus, I recommend the authors attempt to deepen the subject of their article, as the bibliography is too concise. Nonetheless, in my opinion, less than 50 articles for a research paper are insufficient. Currently, authors cite only 33 papers, and in my opinion, they should cite more than it. Therefore, I suggest the authors focus their efforts on researching relevant literature: I believe that adding more citations will help to provide better and more accurate background to this study.

Authors: Thank you for the suggestion of more references needed in the article. We have consulted and cited more references in the revised manuscript.

  1. Introduction:

Write in more detail about the relationship between migraine and fasting/starvation. Why was this investigated?

In addition, it would be worthwhile to write about the role of mitochondrial dysfunction in migraine.

Furthermore, the relationship between inflammatory processes and migraine should be detailed.

At the end of the introduction, do not detail / discuss their results, but write about the aims of their experiment, why they examined the relationship between migraine and fasting, why did they do these studies?

Authors: Thank you for the detailed suggestions. We have revised the part of introduction and have written more to explain the relationship between migraine and fasting, the role of mitochondrial function and inflammation in migraine.

  1. Materials and Methods

In animal experiments, if the behavior of the animals is examined, the minimum number of animals required is 8-10. This number of animals is essential to carry the appropriate statistical test.

Authors: We supplemented the number of animals in the animal experiments. And the number of animals of each group is 12 now.

  1. Figures

Please include the full form of the abbreviations in the figure legends as well.

Figure 1: please include SEM or SD on the graph. 

Figure 3C: n=0.067 - i think they wanted to indicate the value of p. Please check it. 

Authors: Thank you for pointing out the errors. We have revised them in the Figures of the manuscript.

  1. Discussion 

The discussion is unfocused. Only consecutively described sentences, the discussion is not structured properly. 

From the summary, I miss that the authors detail the importance of their results, and it would be good to summarize how their findings relate to knowledge about this disease. 

Authors: We have revised the discussion part of the manuscript.

  1. Language proofreading is recommended.

Authors: We have read through the manuscript and proofread the language.

Reviewer 2 Report

Please clarify the behavioral tests chosen in this study (head-scratching times and climbing cage) as interpreted as headaches in the mice model. Most of the literature points to other types of methods of measurements after induction of the model, for example, head aversion in response to mechanical stimuli by von Frey or thermal stimuli, manually or by orofacial cages (e.g., https://animalab.eu/orofacial-stimulation-test-fehrenbacher-henry-hargreaves-method). Please explain why these methods were not considered.

Please explain why only male mice were used in this study while migraine is 3 times more in females also based on the ARRIVE guidelines, it is highly recommended using of both sexes in basic research

Please add the limitations of this study and justification of the number of animals per group (it was 6?).

In humans, low glucose levels can induce headaches in people without migraine, too. How did the authors explain this? what would be special about migraine brain and metabolism due to fasting? is that lowering the threshold or producing hypersensitivity in response, or prolongation of pain duration or more frequent? Please elaborate on this.

Did the authors test the normality of data to apply parametric tests? In the statistic section, please add more information about this and also related to the software used. If the authors have used mean and SD the expectation s to have normal data or log-transformed to yield normal, but these should be clarified. if data are not normal, then median and IQ should have been used.

Did any complications occur? any death or animals or other safety measures? or all experiments were fulfilled with all subjects in the group?

How these data can be translated and used in terms of human use and clinical practice. Authors are encouraged to give perspectives to their findings and elaborate more on the next step of research.

Author Response

Response to Reviewer 2

  1. Please clarify the behavioral tests chosen in this study (head-scratching times and climbing cage) as interpreted as headaches in the mice model. Most of the literature points to other types of methods of measurements after induction of the model, for example, head aversion in response to mechanical stimuli by von Frey or thermal stimuli, manually or by orofacial cages (e.g., https://animalab.eu/orofacial-stimulation-test-fehrenbacher-henry-hargreaves-method). Please explain why these methods were not considered.

Authors: In our study, we only use head-scratching and climbing cage as the behavioral tests. Because these tests are easy to observe and require no additional equipment. Indeed, they are not very objective comparing with other methods. The method you mentioned above may be better. But unfortunately, we do not have the equipment in our lab. This is a limitation of our study.

  1. Please explain why only male mice were used in this study while migraine is 3 times more in females also based on the ARRIVE guidelines, it is highly recommended using of both sexes in basic research.

Authors: Thank you for the comment. In our study, we focused on energy metabolism disorder in the pathophysiology of migraine and we didn’t use female mice in our study in order to avoid the influence of physiological cycle and hormones.

  1. Please add the limitations of this study and justification of the number of animals per group (it was 6?).

Authors:  We have discussed the limitation of this study in the revised manuscript. And we supplemented the number of animals of each group to 12 in the animal experiments.

  1. In humans, low glucose levels can induce headaches in people without migraine, too. How did the authors explain this? what would be special about migraine brain and metabolism due to fasting? is that lowering the threshold or producing hypersensitivity in response, or prolongation of pain duration or more frequent? Please elaborate on this.

Authors: These are very good questions. We can’t answer all of the questions for the limitation of our study. In the animal experiment, both groups (normal diet and fasting) were administrated with nitroglycerin to make acute migraine model. It may be better to add two groups (normal diet and fasting without nitroglycerin) to distinguish the effects of migraine brain or fasting. In cellular research, we can see that only low glucose also induced more inflammatory factors in astrocytes. This may explain why low glucose levels can induce headaches in people without migraine. About these questions, we have added some content in discussion.

  1. Did the authors test the normality of data to apply parametric tests? In the statistic section, please add more information about this and also related to the software used. If the authors have used mean and SD the expectation s to have normal data or log-transformed to yield normal, but these should be clarified. if data are not normal, then median and IQ should have been used.

Authors: Thank you for pointing out the problems in the statistic section. We have revised them in the manuscript.

  1. Did any complications occur? any death or animals or other safety measures? or all experiments were fulfilled with all subjects in the group?

Authors: No obvious complications occurred and all experiments were fulfilled with all subjects in the group. This modeling method is safe for mice.

  1. How these data can be translated and used in terms of human use and clinical practice. Authors are encouraged to give perspectives to their findings and elaborate more on the next step of research.

Authors: Thank you for the suggestion. We added this in the discussion.

Round 2

Reviewer 1 Report

Dear Authors,

I appreciate that the authors have taken my considerations into account, and all my concerns have been addressed.